# Mild Paravalvular Leak May Pose an Increased Thrombogenic Risk in Transcatheter Aortic Valve Replacement (TAVR) Patients-Insights from Patient Specific In Vitro and In Silico Studies

**DOI:** 10.3390/bioengineering10020188

**Published:** 2023-02-01

**Authors:** Brandon J. Kovarovic, Oren M. Rotman, Puja B. Parikh, Marvin J. Slepian, Danny Bluestein

**Affiliations:** 1Biofluids Research Group, Department of Biomedical Engineering, Stony Brook University, Stony Brook, NY 11794, USA; 2Division of Cardiovascular Medicine, Department of Medicine, Stony Brook University, Stony Brook, NY 11794, USA; 3Department of Medicine, Sarver Heart Center, University of Arizona, Tucson, AZ 85724, USA; 4Department of Biomedical Engineering, College of Engineering, University of Arizona, Tucson, AZ 85721, USA

**Keywords:** transcatheter aortic valve replacement (TAVR), TAV, patient-specific testing, computational fluid dynamics, thrombogenicity, PVL

## Abstract

In recent years, the treatment of aortic stenosis with TAVR has rapidly expanded to younger and lower-risk patients. However, persistent thrombotic events such as stroke and valve thrombosis expose recipients to severe clinical complications that hamper TAVR’s rapid advance. We presented a novel methodology for establishing a link between commonly acceptable mild paravalvular leak (PVL) levels through the device and increased thrombogenic risk. It utilizes in vitro patient-specific TAVR 3D-printed replicas evaluated for hydrodynamic performance. High-resolution µCT scans are used to reconstruct in silico FSI models of these replicas, in which multiple platelet trajectories are studied through the PVL channels to quantify thrombogenicity, showing that those are highly dependent on patient-specific flow conditions within the PVL channels. It demonstrates that platelets have the potential to enter the PVL channels multiple times over successive cardiac cycles, increasing the thrombogenic risk. This cannot be reliably approximated by standard hemodynamic parameters. It highlights the shortcomings of subjectively ranked PVL commonly used in clinical practice by indicating an increased thrombogenic risk in patient cases otherwise classified as mild PVL. It reiterates the need for more rigorous clinical evaluation for properly diagnosing thrombogenic risk in TAVR patients.

## 1. Introduction

Transcatheter aortic valve replacement (TAVR) is a minimally invasive therapy used to treat severe aortic valve stenosis (AS) in patients with calcific aortic valve disease (CAVD). With recent advances in the TAVR procedure and device design generations, TAVR is becoming a standard therapy that is rapidly expanding to younger and lower-risk patients. The basic design of current commercial TAVR devices features a bioprosthetic (chemically fixed tissue) leaflet structure sutured to a stent frame that is deployed over the diseased valve- either by an expanding balloon or self-expandable stented frame. The device is crimped onto a delivery catheter, commonly guided through the aorta from the femoral artery and expanded across the diseased stenotic aortic valve. Since the initial conception of TAVR in 2002 and its first FDA approval in the USA in 2011 [1], TAVR use has grown beyond high-risk patients to include younger and low-risk patients. Partially due to the favorably reduced length of hospital stays, in 2019, the number of TAVR cases exceeded the gold standard surgical aortic valve replacement (SAVR) [2].

With the approval of TAVR in low-risk, younger, and now also bicuspid aortic valve (BAV) patients, TAVR is rapidly becoming the standard therapy to treat AS despite numerous persistent clinical complications. TAVR is prone to various clinical complications, including cardiac conduction abnormalities (CCA), poor TAVR performance due to patient-prosthesis mismatch (PPM), and leakage flows between the prosthesis sleeve/skirt and the lumen termed “paravalvular leak”. These complications have been reduced in severity and prevalence with successive generations of device designs and increased experience of interventionalists. However, thrombosis and thromboembolic events remain persistent. Major stroke rates remain between 1–5.5% [3] in newer generation devices which are reduced from the 7.8% (1 year) rates of the early PARTNER-B trial [4]. Thrombosis, which is often hypothesized to be a result of flow stagnation and unfavorable materials, leads to subclinical leaflet thickening, where the deposition of thrombosis on the aortic leaflet surface causes malfunctioning of the prosthesis. Rates of leaflet thrombosis are greatly varied with each device trial, with rates common rates between 10–15% and some studies reporting rates up to 40% of patients [5,6]. This discrepancy may be due to the lack of symptomatic or the impact on patient outcomes [7], leading to reduced detection rates. With the extension of TAVR to younger patients, rates of subclinical leaflet thickening have been increasing [8,9,10]. Many studies [5,6] have shown rates of leaflet thrombosis are linked to unfavorable TAVR deployment parameters such as eccentric deployments [11] or reduced valve performance due to heavy patient calcification [12], as well as anatomical features such as large sinus of Valsalva leading to increased stagnation [13].

While unfavorable hemodynamics due to stagnation or material surface properties may increase the thrombogenic potential of each device, the risk of thrombosis and stroke due to PVL has not been studied rigorously, and investigations into a possible link have largely been overlooked in clinical trials. PVL leak channels are complex and highly restricted flow paths due to incomplete sealing between the expanded TAVR device and underlying calcified leaflets and the aortic wall that are driven by large diastolic pressure gradients, creating high-velocity jet flows from the native sinuses back into the left ventricular outflow tract (LVOT). PVL is often classified by leak severity determined by clinician judgment of the jet velocity and flow, with newer generation device improvements significantly reducing severe PVL rates in clinical trials (Moderate/Severe PVL at 30 days < 3.5% of patients [14], <0.8% [15]). Mild and trace PVL rates remain common with, for example, rates of no notable regurgitation at 30 days in 19.7% of patients in a recent low-risk trial [14]. PVL severity is often shown to impact many post-operative outcomes and increase patient mortality rates [16,17], which can be attributed to the continued cardiac burden of often high-risk patients. An abstract by Rahgozar et al. showed no link between major stroke rates and the classification of PVL [18]; however, larger studies have contradicted these findings. Padang et al. [19] demonstrated that mild and tract PVL had a lower survival rate (50.9%) compared to no PVL (62.7%) at 5 years. In a recent study, Saito et al. [20] showed that mild or greater PVL had significantly lower freedom from events (70 months) compared to trace and no PVL. Additionally, PVL severity has been shown to be related to hemolysis rates [21].

The link between thrombosis in TAVR and PVL was initially investigated in several in silico computational fluid dynamics (CFD) simulations [22,23,24,25]. PVL has been replicated with a patient-specific in vitro flow model, giving the ability for further in-depth study of the clinical scenarios for device comparison [26,27,28]. The in silico models have relied on initial structural finite element simulations to estimate deployment of the TAVR device in the patient anatomy, and the resulting PVL channels could only be compared to echo/Doppler imaging which is limited in resolution and accuracy of flow velocity [23,24,26]. These studies have utilized device thrombogenic emulation (DTE) methodology in which a particle model is used to estimate platelet trajectories, and stress accumulation along the trajectory is collapsed into a probability of device thrombogenicity or the device thrombogenic footprint [29,30,31,32]. In these studies, platelets were seeded in the blood flow field during diastole, and their individual trajectories were tracked over a single wash through the channels–indicating a significant increase in the device thrombogenic potential due to the leak [23,24,25].

In this study, we presented an evolution to the DTE methodology combined with an innovative approach of generating patient-specific in silico models from the in vitro models that were reconstructed from in vivo CT scans of patients. The patient-specific in vitro replicas were used to evaluate the valve hydrodynamics of the deployed TAVR device under conditions that closely mimic those of the in vivo deployed device and used to validate the resulting in silico flow results. The models with the deployed TAVR device were high-resolution µCT scanned and reconstructed into in silico models. The DTE methodology was expanded by increasing the amount of platelet-like particles tracked, as well as tracking them in successive cardiac cycles. We demonstrated the increased thrombogenic risk due to mild PVL flows that may otherwise be considered clinically acceptable. We conducted an extensive analysis of the complex platelet trajectories generated by the flow patterns within the PVL channels.

## 2. Materials and Methods

### 2.1. In Vitro Model Creation and Hydrodynamic Testing

In a previous study [27], elastomeric patient-specific CAVD replicas and matching silicone aortic arch were created in order to evaluate the performance of different TAVR devices in a more clinically relevant in vitro hydrodynamic environment. Five patient anatomies were reconstructed from cardiac CT scans of (de-identified) patients (Stony Brook University Hospital, IRB approval 2013-2357-R5) and were selected based on post-TAVR clinical complications, large range of valvular calcific masses, and aortic root and left ventricular outflow tract (LVOT) anatomical morphologies. The patient models and deployed TAVR devices were tested in the Mentice Inc. (Stony Brook, NY, USA) Replicator Pro simulator [27,33,34] (Figure 1) and evaluated according to ISO 5840-3 hydrodynamic standard [35].

The five CAVD replicas were used in this study along with a unique polymeric TAVR device, PolyV-1 (20 mm, PolyNova Cardiovascular, Stony Brook, NY, USA). The deployed valve was evaluated within each anatomy at a cardiac output (CO) of around 5 L/min at 70 BPM under normotensive aortic conditions (120/80 mmHg). The performance of the device is quantified during forward flow with targets of the average systolic pressure gradient (ΔP, mmHg) and effective orifice area (EOA, must ≥0.85 cm^2^ at 5 L/min, Equation (1)), which is calculated in units of cm^2^. $Q_{\mathrm{RMS},\mathrm{systole}}$ is the root mean square of the systolic forward flow (ml/s), and ρ is the density of the working fluid (g/cm^3^).
(1)$$\mathrm{EOA}=\frac{Q_{\mathrm{RMS},\mathrm{systole}}}{51.6\;\sqrt{\frac{{\Delta P}_{\mathrm{avg},\mathrm{sys}}}{\rho }}}\;\mathrm{and}\;\mathrm{EROA}=\frac{Q_{\mathrm{RMS},\mathrm{diastole}}}{51.6\;\sqrt{\frac{{\Delta P}_{\mathrm{avg},\mathrm{dia}}}{\rho }}}$$

Diastolic or backflow performance is quantified with regurgitant flow targets of paravalvular leak (PVL) flow (≤10% SV) and a total regurgitant fraction (RF, ≤15% of stroke volume (SV), combination of closing and leak flows). Equation (1) also features a modified version of EOA- termed effective regurgitant orifice area, which uses the same constants and assumptions to estimate the regurgitant opening area. The in vitro performance of PolyV-1 in the five anatomies at 5 L/min varied because of the interaction between the device and the CAVD replica, leading to eccentric deployments. Although the valve performance was varied, the valve exceeded the ISO-required targets, and all performance targets would be indicative of a successful clinical outcome. Classification of the PVL flows shows that Anatomy A, C, and E represent mild/moderate flows, and B and D represent mild flows-likely leading to a low level of clinical concern.

### 2.2. CT and Geometry Reconstruction

After collecting the hydrodynamic performance, the CAVD replica with the deployed PolyV-1 was carefully removed from the Replicator Pro simulator and placed in a vessel and jig designed to hold the model centered and still during µCT scanning. The silicone aorta was not utilized due to bore size limitations of the scanner. The vessel was filled with a 20% mixture of contrast dye (Omnipaque 240, GE Healthcare, Chicago, IL, USA) and was initially placed in a vacuum chamber to remove bubbles. The contrast dye helped flood the PVL channels, allowing visualization, and distinguishing between the anatomical features of the replica and the polymeric sleeve of the device (Figure 1). The models were scanned (VivaCT 45, SCANCO USA Inc., Wayne, PA, USA) at 17 µm voxel resolution.

The resultant images were then processed with a MATLAB (Mathworks, Natick, MA, USA) code to filter, smooth, and enhance the images to reduce the noise and scatter from the metallic stent. The processed images were used with ScanIP software (Synopsys Inc., Mountain View, CA, USA) to reconstruct the deformed native leaflet and LVOT geometry, the stent frame, and the polymeric sleeve. The TAVR leaflets geometry was recreated in the closed position (Rhinoceros 3D, Robert McNeel & Associates, Seattle, WA, USA), utilizing landmarks from the µCT scan for locations of attachment to the polymeric sleeve. The LVOT and native sinuses were joined to the original geometry of the patient-specific replica in order to complete the CFD domain. Algorithmic geometry smoothing was performed during the reconstruction, and manual smoothing (Meshmixer, Autodesk Inc., San Rafael, CA, USA) was done on the connection of the geometry to reduce any sharp features in the fluid domain. In order to reduce the scan size and data, the crown region of the stent was not scanned and reconstructed.

### 2.3. CFD Model Setup

The reconstructed models were meshed in Fluent v20R1 (ANSYS Inc., Canonsburg, PA, USA) using polyhedral cells. Sizing functions followed similar studies [23,24,25,26], and additional mesh and temporal convergence were confirmed with cell numbers between 2.9 and 2.5 million cells for the models as well as a constant 0.5 ms fixed timestep. Laminar flow solution was assumed, following previous CFD studies of PVL flows [23,24,26,36], and confirmed by calculating the average Reynolds number along the streamlines during peak diastole between 115–236 and instantaneous maximum Re between 990–1917 (See Appendix A). Inlet/outlet extensions (6 cm) were added from the LVOT to the ventricular inlet and from the ascending aorta to the aortic outlet.

Pressure gradient boundary conditions (BC) were obtained from the in vitro testing, and the gradient waveform was assigned to the ventricle inlet. In order to capture systolic and diastolic flow, the TAVR leaflets were meshed and assigned with Darcy’s law governing the porosity model. The inverse permeability (D) of the leaflets was varied to be “open” or permeable (D = 0 m^−1^) during systole and “closed” or impermeable (D > 10^10^ m^−1^) during diastole. The goal of this model is to capture the bulk flow of the valve without the additional complexity of fluid-structure interacting (FSI) modeling of the leaflet motion [25,37] (as well as avoiding tracking errors in the particle model). The blood was modeled as a Newtonian fluid with density and viscosity matching the glycerol solution of the in vitro blood analog (50.3% glycerin [27,33,34] at 37 °C, µ_dynamic_= 3.5 mPa s, ρ = 1120 kg/m^3^).

Each model yields unique flow patterns, seen in Figure 2, and accurate matching of the bulk flow waveforms between the in silico and in vitro results. A further agreement between the in silico and in vitro results is seen in the t-test results in Table 1. There are no statistical differences between the two groups regarding the performance characteristics of the bulk flows. The largest discrepancy is seen in Anatomy C, where the largest pressure gradient was not able to drive the same forward flow as in the in vitro model. However, this is an extreme case with a highly irregular/eccentric deployment. The flow was simulated for 5 continuous cycles at 70 BPM.

### 2.4. DPM Modelling Platelet-like Platelets

Characterization of the thrombogenic potential of the PVL flows is determined with a discrete phase model (DPM), previously used by our group to evaluate various cardiovascular devices [23,24,25,29,30,31,32]. The DPM method seeds neutrally buoyant spherical particles used to represent the platelets (∅ = 3 µm) and compute the Lagrangian trajectories (accounting for drag and particle momentum) with the two-phase interaction, with the continuous flow domain creating a fluid-structure interaction (FSI) simulation. Platelets were seeded 2 cm proximal from the bottom (ventricular) stent frame and were equally spaced 200 µm apart across the transverse plane. Seeding began at the start of the third cycle (after flow periodicity was established) and was seeded each 0.5 ms timestep for 210 ms (the entire positive systolic pressure gradient). Each simulation took 10–12 days to solve all five cardiac cycles with a 32-core AMD Threadripper 3970X processor workstation.

The instantaneous particle path location, particle velocity, and scalar stress value were exported for every 2 timesteps for the remaining 3 cycles. The scalar stress value (Equation (2)) reduces the shear and principal stresses on the particle into a scalar value and is used in calculating the stress accumulation on each particle (SA, Equation (3)) [32,38]. The thrombogenic potential of each anatomy/flow is compared by collapsing all the particle path stress accumulations along each trajectory into a probability density function (PDF) to compare the likelihood of higher stress flows experienced by the platelets [39]. Additionally, the platelets were tracked along each trajectory and separated into platelets that entered the PVL channels and those that passed inside the TAVR valve and washed into the aorta.
(2)$$\sigma =\sqrt{\frac{\sigma_{\mathrm{xx}}^{2}+\sigma_{\mathrm{yy}}^{2}+\sigma_{\mathrm{zz}}^{2}-\sigma_{\mathrm{xx}}\sigma_{\mathrm{yy}}-\sigma_{\mathrm{yy}}\sigma_{\mathrm{zz}}-\sigma_{\mathrm{zz}}\sigma_{\mathrm{xx}}+3\left(\tau_{\mathrm{xy}}^{2}+\tau_{\mathrm{yz}}^{2}+\tau_{\mathrm{zx}}^{2}\right)}{3}}$$
(3)$$\mathrm{SA}=\sigma \cdot t_{\exp}=\overset{t_{\exp}}{\underset{t_{0}}{\int }}\sigma \left(t\right)\mathrm{dt}\;\approx \overset{N}{\underset{i=1}{\sum }}\sigma_{i}\cdot \Delta t$$

## 3. Results

Analysis of the flow patterns, velocities, and stresses along multiple platelet trajectories revealed complex flow dynamics emerging from specific patient anatomies that increase the thrombogenic potential of the device. Each anatomy showed evidence of a large number of platelets flowing into the PVL channels in both systolic and diastolic phases of the cardiac cycle (an example of such flows in specific anatomy and PVL channel formed after the TAVR deployment is shown in Figure 3). The platelets additionally may enter the channel flow multiple times throughout the cycles, with some platelets entering and leaving up to six times during the three cycles. The bottom section of Table 1 shows that the range of platelets entering the PVL domain varied from 6.5% to 23.2% for the given target cardiac output. Between 1.5% and 8.1% of platelets re-entered the PVL domain (platelets had to exit the PVL domain at least once before being tracked again into the domain). Figure 3 highlights an example path of a platelet in Anatomy C chosen randomly to highlight such an occurrence in which a platelet enters the PVL flows multiple times. This was not a rare occurrence given the large number of seeded platelets. It illustrates that the platelet remained in the PVL channel during the second cycle before being entrained into the flow in the third cycle, demonstrating non-consecutive particle entrainment in the pulsatile PVL flows. Additional complex particle trajectories can be seen in the Appendix A.

The platelet trajectories were tracked for the stress accumulation over the trajectories and sorted for visualization purposes, with Figure 4 depicting the pathlines of the top 3000 platelets with the highest stress accumulation (SA) values, i.e., those with elevated thrombogenic potential. Additional cartesian views and full videos of 1 M randomly sampled platelet trajectories can be seen in the Appendix A and Appendix A. Figure 4 demonstrates a critical feature of PVL thrombogenicity when comparing the left-column platelet velocities to the right-column stress magnitudes, in that areas of high-velocity flows are not directly associated with high-stress regions. Large velocity channels in Anatomies B and E do not have corresponding high-stress regions compared to the higher-stress regions of Anatomies A and D. Observing the channel paths and platelet trajectories show convergent and divergent sections where the platelets can accelerate within the narrowing (mimicking stenosis-like condition). This acceleration, with maximum velocities reaching around 4 m/s within a narrowing, produces increased shear stresses on the platelets- resulting in stress accumulation likely to activate the platelets.

### 3.1. Thrombogenic Footprint

The platelet trajectory stress accumulations results were collapsed into the probability density functions (PDF) seen in Figure 5—the device thrombogenicity emulation (DTE) methodology [29,30,31,32]. The PDF charts compare each anatomy and are separated into two curves of the DPM platelets that entered the PVL domain (blue) and those that did not enter (orange) or remained/washed into the aortic flow. The thrombogenic footprint of a device is interpreted as PDF curves with the higher/right shift tail of the curve representing the larger stress accumulation values—corresponding to a larger potential for platelet activation and possible thrombogenic response. While many platelet activation models exist and there is no common consensus, a commonly accepted threshold for stress accumulation that may activate the platelets is the Hellums’ criteria of 35 dynes × s/cm^2^ or 3.5 Pa × s [40]. Platelets experiencing a combination of stresses and exposure to them above this threshold are more likely to activate. While the PDF technique utilizes bootstrapping statistics to allow the comparison of two systems with differences between the number of platelet trajectories, e.g., the number of total platelets versus the PVL platelets, it was critical to seed the simulation with enough platelets to fully capture the device thrombogenic potential. This can be seen in the even distributions of each curve in the PDF, with complete lower tails (even range of SA values and sampling). The peaks of the orange curves indicate that each anatomy had similar stress values in the forward flow between the TAVR leaflets or that there is minimum stress accumulation resulting from the diverse valve performance range.

The PVL platelet curve (blue) in Figure 5 demonstrates that in all anatomies, there was a prominent increase in platelet stresses within the PVL channels. In all Anatomies except E, the prominent peak of the PVL platelets overlapped Hellums’ criteria, with notable right-skewed distributions in Anatomy A, C, and D. Anatomy E appears to have two peaks (bimodal) in the stress accumulations, which is most likely due to specific flow patterns and the highest stress platelet locations (Figure 4)- not corresponding to the location of the highest velocity platelet locations. The largest stress-inducing flow regions in Anatomy E appear to come from smaller numerous leak paths and at the base of the stent meeting the LVOT lumen. The prominent leak path between the right and non-coronary leaflets has the largest velocity but the low-stress magnitude, which contributes to recirculating more PVL platelets without accumulating significant stresses. Figure 5 also contains inset figures expanding on the PDF above 5 Pa × s to show that there are no tail regions of extremely high-stress accumulation platelets, with the orange curve and the blue curve demonstrating increased thrombogenic potential at the higher accumulation values. SA values of the PVL platelets were analyzed for mean values to demonstrate that Anatomies A and D appear to have the largest thrombogenic potential- with respective mean SA of 3.75 and 3.29 Pa × s, and Anatomy E with the lowest mean 1.04 Pa × s.

### 3.2. Platelet Trajectory Characteristics and Effects

One of the goals of the large number of platelets seeded in these simulations was to reduce the bias or the effect of a single timestep or the choice of the plane for platelet seeding that was traditionally done in previous DTE simulations. The comparison of the platelet PDFs based on the time period of the injection time (figures can be seen in Appendix A) showed that platelets will have various SA values based on the injection time. There was no clear trend among these five anatomies, with some simulations having larger SA values from platelets injected at early systole and other anatomies having larger SA values at end-systole. It is evident that the approach utilized in this simulation is necessary to avoid biasing the simulation results.

The next effect that was analyzed was comparing the starting location of the PVL platelets within the LVOT. The contour plots on the bottom of Figure 6 demonstrate the percentage/concentration of platelets based on the starting location of the platelets (over all the injection timesteps) that enter the PVL channels. It is evident that in all anatomies, PVL platelets tended to reside near the lumen of the LVOT as compared to the center flow. The contour plots show hotspots near the lumen, and the center of the contours indicates that only a few platelets enter the PVL channels from the central flow. The top contour plots in Figure 6 demonstrate the PVL velocities (CFD) exiting the valve into the LVOT and point to possible clinical measurements that correspond to such (Echo/Doppler or magnetic resonance angiography). The starting location of PVL platelets does not appear to be influenced by the location of the PVL channel exit or velocity. Therefore, it is not evident that entering the PVL channel during the first systolic period impacts the overall phenomena.

## 4. Discussion

The study of five anatomies presented here with mild/moderate PVL flows strongly indicates that there is clear evidence of increased thrombogenic potential attributed to the PVL flows and that in specific CAVD patients treated with TAVR platelets enter PVL channel flows repeatedly over many successive cycles. This strongly suggests the need for and importance of studying trends or effects of the hemodynamics indicative of increased thrombogenic potential in what may otherwise be diagnosed as mild PVL, presently largely disregarded as conveying clinical risk for the TAVR patient. While acknowledging that in our study these trends are limited by the small sample size of five demanding simulations, nevertheless, those were carefully chosen to represent a range of typical cases that were labeled as mild PVL. Simple Pearson linear correlation coefficients (r) were initially investigated by comparing the hydrodynamic parameters and the resultant DPM platelet data (percentage of PVL platelets and stress accumulation data). The results of these correlations can be seen in the Appendix A. It is hypothesized that leak flow rates or EROA values would correlate with the number of platelets entering the PVL flows; however, in these results, there is only a minor relation with coefficients of 0.42 and −0.43 for the respective EROA vs. % of PVL platelets and leak flow vs. % PVL platelets. EROA vs. Mean SA values and Leak Flow vs. Mean SA values seem to have a larger effect with respective coefficients of 0.68 and −0.84. However, observing the distribution of data and resulting trends indicate that these correlations are limited by the sample size.

Further analysis of the flow field and platelet dispersion patterns beyond the simple bulk flow measurements was needed to find clear trends and influences of the PVL platelet trajectories and thrombogenic potential. A counterintuitive phenomenon was initially noted when observing the platelet trajectories, where lower-performing valves (smaller EOA) resulted in a higher valve jet velocity during systole, resulting in the platelets being cast further downstream away from the PVL channels. This was also dependent on the aortic arch structure and jet direction, but it was observed that the leak flow was not the only major factor in determining the percentage of platelets entering the PVL flow channels. The ratio of EOA/EROA [cm^2^/cm^2^] stemmed from a hypothesis that the incorporation of platelets must depend on the forward flow performance of the valve. The ratio is a measure of the forward performance over the leak performance of the valve in an attempt to capture how well the platelets fill the sinuses during systole or are cast downstream over a measure of how much leak flow was generated given the PVL flow rate. The RF (percentage of total leak flow/ stroke volume), in theory, would capture the degree of forward and back flows, but RF is highly dependent on the cardiac output (Table 1) and does not measure the degree of valvular stenosis. EOA and EROA measures help account for the stroke volume and the time period of forward flow (Q_rms_, Equation (1)), helping to negate CO discrepancies. It is important to note that both EOA and EROA can/are routinely captured with clinical echo/Doppler and should be investigated in future in vivo studies for links to thromboembolic events. Figure 7 top row indicates a trend between the ratio of EOA to EROA and the percentage of platelets entering and re-entering the PVL flows (r^2^ values of 0.75 and 0.82, respectively). As a point of comparison, the regurgitation fraction (RF), which is defined as a percentage of the stroke volume that flows backward, did not capture this trend (r^2^ values of 0.30 and 0.56 for RF vs. % PVL platelets and % re-entering platelets). Anatomy B had the lowest RF of 6.6 %SV and the lowest percentage of PVL platelets at 6.5%. In contrast, Anatomy A had the largest RF of 20.2 %SV and the second least percentage of PVL platelets at 14.7%. The other three anatomies had a similar percentage of PVL platelets (between 22.3–23.2%), with a wide range of RF (13.1–18 %SV). Lastly, in Figure 7, to the right of the top row, there is a relation between the number of PVL platelets and the number of platelets re-entering.

In order to study further the effects on the thrombogenic potential, beyond observing the bulk leak flow data yet maintaining typical CFD parameters, both the diastolic streamlines (Appendix A) and exit PVL velocities were studied. While the platelet simulations quantify the most accurate depiction of the device thrombogenic footprint, it is useful to find CFD parameters that may help expand or reinforce the thrombogenic potential while keeping a reduced computational burden. In Figure 7, the bottom row shows the mean and median SA values for the PVL platelet trajectories versus such standard CFD parameters. The peak diastolic streamlines (seeded above the valve) velocities were averaged and demonstrated a clear trend linking the thrombogenic potential and PVL velocities (r^2^ = 0.75 mean and r^2^ = 0.83 median SA values). The diastolic streamlines exclude the intricacies of the PVL flows and channels, e.g., the length of the platelet path or the shear stresses within the flow.

Since streamlines can only be determined with CFD simulations, and current imaging techniques are not able to accurately determine the jet velocity in the PVL channels, another goal was to determine if an additional imaging plane could aid in the clinical determination of PVL thrombogenicity. The PVL velocity contour plots (LVOT) in Figure 6 were generated as an alternative imaging plane that could be acquired with Echo/Doppler or MRA imaging. These contours represent a single plane as close to the base of the TAVR device that could be imaged and measured the PVL jet velocities. These velocities were filtered to remove stagnant flow patterns (<0.3 m/s), and the jet velocities were averaged and compared to the thrombogenic potential (Figure 7 bottom row). The mean and median SA values compared to the average PVL exit velocity with respective r^2^ values of 0.65 and 0.7. These trends are weaker compared to the streamlined CFD data but could be utilized in the clinic to aid in future studies of TAVR thrombogenicity instead of the imprecise and potentially biased PVL ranking system utilized in current clinical diagnosis.

### 4.1. Study Significance

The main impact of this study is the demonstration of the value and utility of observing the influence that even mild PVL flows have in order to anticipate clinical thrombotic events. To our knowledge, this is the first study to demonstrate that there is a clear thrombogenic potential arising from mild and mild/moderate PVL flows, independent of the leak volume alone instead of resulting from the complex PVL channel morphological characteristics that could only be studied with an in-depth in silico study. The current clinical classification of PVL depends on inaccurate echo/Doppler measurements of flow velocity/rate or observation of only the prominent PVL channels; these measurements cannot capture the flow patterns in TAVR devices deployed in patient-specific anatomies and the resulting risk from these flows, as demonstrated by our careful study and analysis. Platelets experience increased shear stresses in the PVL flows due to the high-velocity flows and elevated shear stresses and large diastolic pressure gradients, which results in higher stress accumulation on the platelets as compared to platelets flowing in the central forward flow. The PVL channels have large and complex pathways that are not easily resolved or imaged and are critical for the assessment of the thrombogenic potential of these mild PVL ranked flows, strongly indicating why this computational assessment is critical to understanding the clinical risk.

Along with potential increased stress accumulation of platelets, this study indicated an increased probability of platelets entering the PVL flows multiple times. For the sake of simplicity, we termed this phenomenon *PVL platelet entrainment*—describing the recruitment of platelets into the PVL jet flows. To our knowledge, this phenomenon has not been researched nor acknowledged as a source of increasing thrombogenic potential following TAVR procedures. The depth and complexity of the present study are not warranted in clinical diagnosis practice. Regardless, it is fair to assume that the in silico observed entrainment is a faithful presentation of the physiological domain and may also be present with other forms of leak flows (paravalvular and intravalvular). A critical finding stemming from the starting location contour plots in Figure 6 is that we can hypothesize that since platelets flowing closer to the aorta’s wall are more likely to be entrained in the PVL flow and that the PVL jets (Figure 6 top row) also exit near the lumen of the LVOT, this entrainment may reinforce the re-entry of platelets into the PVL channels observed in our study. If the platelets exit the PVL channels near the wall, they would be more likely to be re-entrained into the PVL flow channels on subsequent cardiac cycles, thus increasing the stress accumulation and the likelihood of platelet activation.

The novel methodology in this study, with the PVL channels replicated and evaluated in vitro patient-specific model with high-resolution µCT scanning leading to the reconstruction of a complex and accurate CFD domain, provides a set of bona fide simulations with excellent agreement between the in vitro and the in silico flow measurements. The distinct advantage of performing in vitro hydrodynamics prior to incorporation into in silico simulations is in warranting a validation by comparing the resulting flows in both models. Both in vitro and in silico models rely on assumptions of the tissue material properties and are arguably more critical in patient-specific simulations. The standard methodology for patient-specific TAVR simulations in literature has relied on an initial structural simulation followed by the CFD simulation [22,23,25,26]. For example, it could, at most, utilize low-resolution clinical imaging for validation of the PVL location. The agreement between the leak flow volumes in the in vitro and in silico simulations reiterates that the reconstructed PVL channels from the µCT scans were accurate and that the ground truth of the in vitro flow physics corresponds very well to the flow results predicted by the numerical simulations. While the in silico simulation presents a high degree of complexity, currently, it is the only option to obtain detailed physiological estimations of platelet trajectories and the resultant thrombogenic potential.

The novel leaflet porosity model was an effective method to replicate the entire cardiac cycle in CFD without the unwarranted complexities of comprehensive FSI modeling of the leaflet motion. The progressive increase in the internal resistance across the leaflets aided in numerical stability and did not require any adaptations for the DPM platelet FSI model within the ANSYS Fluent code. During the closed valve diastolic period, the flow across the leaflet interface was calculated for each model and confirmed to be less than 0.1 mL/beat, which was assumed to be negligible. The porosity-modeled leaflets replicated the effect of closed/open leaflets on the bulk flow through the TAVR valve and provided a continuous CFD domain. The results of this study do not make any assumptions about the leaflet motion nor the impact on the platelet stresses (discussed in the Limitations below). Having acknowledged the assumptions of the porosity model, the benefits of reduced simulation complexity with CFD modeling of the flow domain allowed the expansion of the DPM model.

The DPM platelet model in our study is one of the largest simulations performed in terms of the number of platelets and in most comprehensive in terms of unbiased seeding and duration while using the DTE method. With the increase in available computational resources, this expansion of the DTE method enabled us to introduce a more realistic seeding pattern and duration, necessarily increasing computational costs. The goal was to seed the platelets in a homogenous distribution pattern, approximating the mixed blood properties while blood is ejected from the left ventricle-seeding the platelets during the entire systolic period in order to avoid seeding time-induced variations that may affect the resultant thrombogenicity. With the presented seeding method, the number of platelets varied from anatomy to anatomy based on the area of the LVOT plane, but the large number of platelets seeded in each anatomy provided the desired accuracy of the probability density function while enhancing the validity of the observed platelets entrainment phenomenon.

### 4.2. Clinical Significance

In this study, we conducted a comprehensive and rigorous patient-specific in silico and in vitro analysis of a cohort of TAVR patients classified as having mild PVL, which demonstrates that their ranking as mild PVL-otherwise assumed not to carry a risk of thrombus formation and stroke, likely underestimates the actual thrombogenic risk that these patients are exposed to. Specifically, our study demonstrates that in defined CAVD anatomies, platelets entering the PVL channels during systole may also re-enter the PVL channels multiple times in successive cardiac cycles. The percentage of platelets entering the PVL flows was highly varied. Opposite to the common hypothesis/assumption that larger leak flows would entrain more platelets, it does not appear to be strongly correlated to the leak flowrate alone. It revealed that there is a complex interplay of flow patterns in the TAVR valve domain that may significantly increase the thrombogenic risk even in mild PVL cases. There is a direct relationship between the percentage of PVL platelets and the percentage of PVL re-entering platelets.

The DTE methodology is comprehensive in quantifying a device thrombogenic potential and the influence of clinical parameters on this potential. It is a computationally demanding simulation that may not be easily expanded to a large cohort of patient cases. It is, therefore, critical to find the key parameters that can elucidate the resultant thrombogenic potential. Relationships between less fine-grained methods such as bulk hemodynamic parameters (EOA, EROA, leak flow, RF) and the average/median stress accumulation values appear to mask this thrombogenic risk as it showed no clear or strong correlations to the thrombogenic potential. Limitations of current clinical imaging (spatial resolution, scatter, and accurate velocity assessment) preclude observing the complex PVL channels and flow patterns within them. It is not possible to replicate the details revealed by our extensive CFD analysis, yet it appears crucial in order to determine the actual thrombogenic potential of the device. Given that in our study, we only used a small set of patient cases representing a range of differing PVL flow channels that would rank as mild. To offer a simpler practical approach that may be better suited for the current clinical practice, we proposed a clinically relevant measurement of the average PVL jet exit velocity proximal to the valve, which in this study had a relationship to the thrombogenic potential. MRA imaging of the transverse plane beneath the TAVR valve could accurately measure the number and jet velocities of the exiting PVL channels for this measurement. This study demonstrated the need for future clinical investigations of the link between even mild PVL flows and thrombogenic events and emphasized the need for the expansion of imaging protocols/methodologies and hemodynamic analysis.

### 4.3. Limitations and Future Directions

In this study, we assumed a clinical relevance/accuracy of the results of patient-specific in vitro replicas that were previously studied and published [27], demonstrating that they represent far better the in vivo conditions for testing the hydrodynamic performance of TAVR devices. In the current study, the valve design utilized, while representative of clinical TAVR, is investigational and not in clinical yet. Based on accepted valve testing standards (ISO 5840:1–3 2021) [35], the resultant hydrodynamic waveforms, valve performance, and leak flows are appropriate. The in vitro to in silico adaptation methodology may appear removed from the in vivo sequelae. However, it offers the advantage that the physiological performance results of the valve can be confirmed or verified between different model results. A limitation of the reconstruction methodology is the large degree of manual segmentation, cleaning, and smoothing of the models due to the complexity of µCT scanning. The segmentation process is manual and computationally burdensome given the large dataset; this process requires in-depth knowledge of the valve structure and anatomy as well as image compression, contrast adjustment, and sharpening to distinguish distinct features. Additionally, the TAVR leaflets needed to be approximated into a closed state because the thin leaflets were barely captured in the µCT scan.

The small sample size of five patient models with one investigational TAVR device is a potential limitation. As mentioned earlier, this was necessitated by the complexity and comprehensiveness of our in vitro to in silico methodology. However, these five cases do represent a range of various otherwise classified patient-specific mild PVL cases. This study serves as a methodological approach demonstrating that the thrombogenic potential can be obtained in these models. It is presented as a proof-of-concept study to be further confirmed with future studies. The authors acknowledged that the correlations in this study are limited by the small set size and range of mild and mild/moderate PVL flows studied. The in-depth analysis of the results reiterated the need to find relevant and discerning factors impacting the thrombogenic potential that can be translated to clinical practice. Another limitation of the study is proper validation with in vitro platelet activation experiments of these patient-specific models. This is currently not possible with the limitation of the materials used (polyurethane CAVD replicas need additional testing for inherent material thrombogenicity and large fluid volume of the Replicator system). We intended to address this limitation in future studies.

The CFD model assumes laminar flow conditions, which have been utilized in previous PVL studies [23,25,26,36], with the mesh refinement focused on the PVL channels. The DTE thrombogenicity model averages and renders the instantaneous components of the stress tensor that the platelets are exposed to into a single scalar value which is used for computing the stress accumulation (SA) along the platelet trajectory. Viscosity and density assumptions of the working fluid approximated the influence of red blood cells (RBC) on the bulk flow, but the platelets would locally experience the viscosity of blood plasma. Approximating the blood flow as a Newtonian continuum with RBCs was necessary to capture the accurate hemodynamics of the macro-scale domain (TAVR valve diameter 20 mm compared to the 3 µm platelet diameter). It is beyond the scope of the current study to include RBC influence. It is assumed that the RBCs are excluded from the PVL flow channels. Additionally, the authors acknowledge replacing an FSI model with an equivalent porosity leaflet model, which does not capture the stresses and motion of platelets interacting with the leaflet motion. The leaflet tip motions are dynamic areas with increased shear stresses and can be sources of thrombogenicity, which are not captured in this study. We justified this assumption by the fact that the shear stresses within the PVL flow channels are larger and offer a much larger contribution to the thrombogenic potential of the TAVR device.

## 5. Conclusions

In the present study, we developed a novel methodology to investigate the link between thrombogenic risk and what is otherwise regarded as a clinically acceptable mild paravalvular leak in TAVR. This methodology generated in silico models from reconstructed high-resolution µCT scans of patient-specific in vitro replicas. Flow validation for the in silico models is provided by the hydrodynamic evaluation of the in vitro models. This study expanded the Device Thrombogenicity Emulation (DTE) methodology, which was previously established to determine the thrombogenic footprint of a cardiac device by drastically increasing the number of platelets tracked, here used to estimate the increased stress accumulation on platelets induced by PVL channels formed after TAVR deployment in patients, to better represent the biology of blood flow via these channels. The significant increase in the number of platelets seeded and analyzed comes at an increased computational cost. However, it brought the simulations closer to the biological ground truth and aided in generating a fair assessment of platelet response to the elevated stresses in the PVL channels over multiple cardiac cycles that clearly indicated an increased thrombogenic potential resulting from mild PVL flows. It demonstrates that this thrombogenic potential is dependent on patient-specific PVL channel dimensions and morphologies rather than the present clinical classification of PVL. This study additionally highlighted complex platelet trajectories with platelets entering the PVL flows multiple times during successive systolic and diastolic cycles that further contribute to the thrombogenic potential. Platelet entrainment into the PVL flow channels was established to be a function of the degree of forward and back flow performance of the devices. The findings of this study demonstrated that further investigation of mild PVL flows, both in the clinic and in future in silico models, needs to be conducted. Our findings reinforced that such higher-resolution methods are of great utility in evaluating the actual thrombogenic risk of TAVR patients.

## Figures and Tables

**Figure 1 bioengineering-10-00188-f001:**
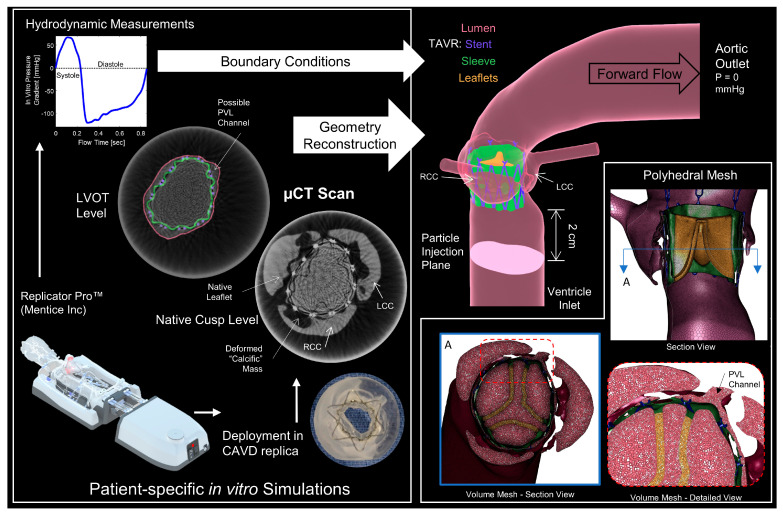
Creation of CFD domain and mesh from µCT scans of in vitro models—Left box—the process of obtaining patient-specific boundary conditions from hydrodynamic studies and segmentation of µCT models (colored to show components). Top right—final sample reconstructed CFD domain. Bottom right box—Polyhedral mesh showing the resolution within the PVL channels and cell volume distinction.

**Figure 2 bioengineering-10-00188-f002:**
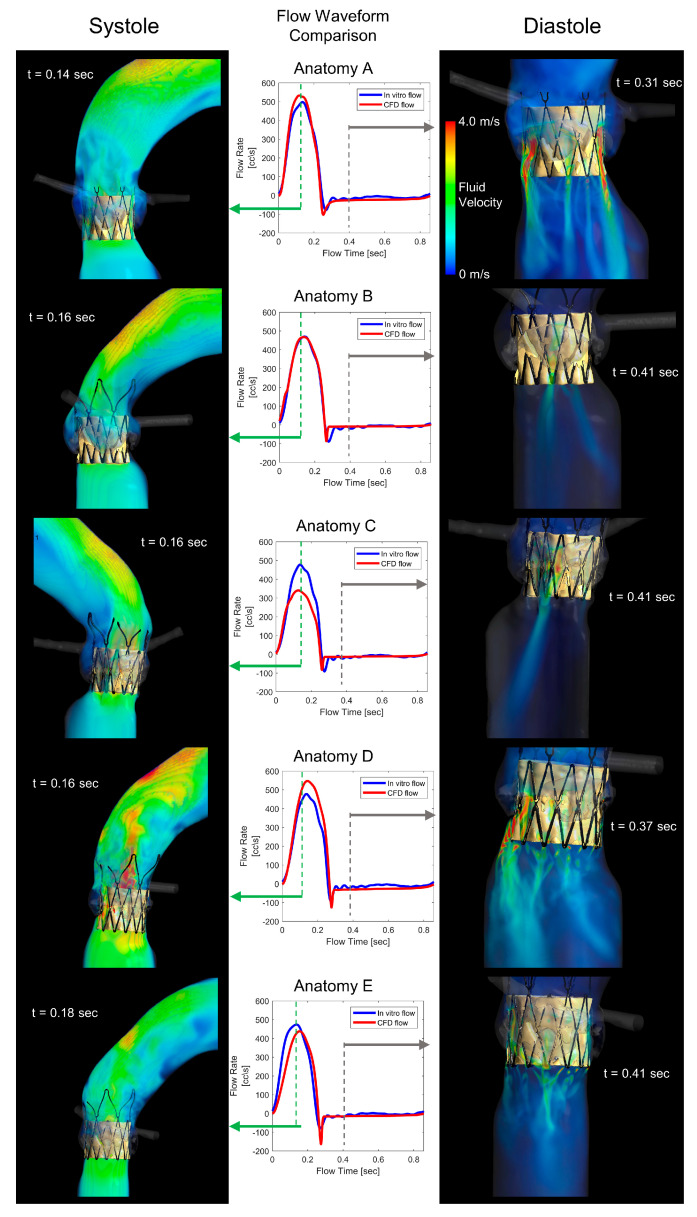
Flow pattern and waveform comparison—Volume of fluid (VOF) rendering of the fluid velocity of the CFD solution in systole (**left**) and diastole (**right**). Overlays of the flow waveform, comparing the in vitro and in silico solutions, with systole (green) and diastole (gray) timepoints labeled.

**Figure 3 bioengineering-10-00188-f003:**
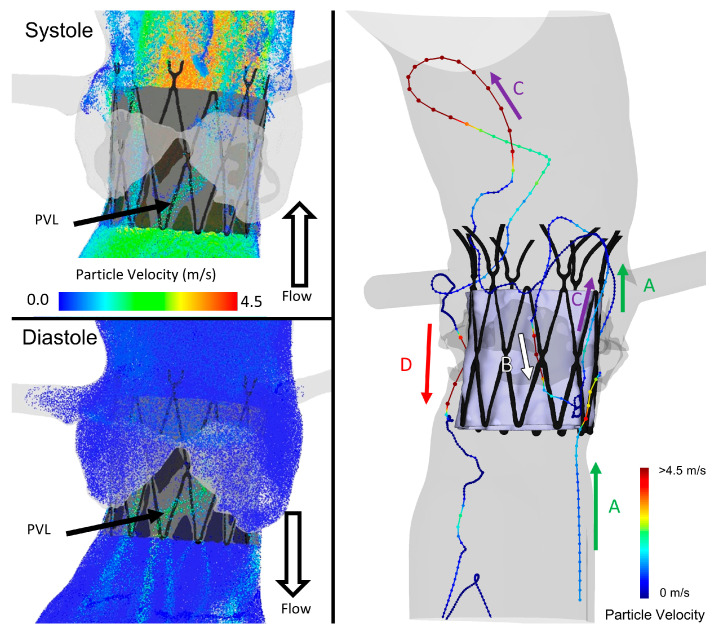
Evidence of platelet entering and reentering in both flow phases—Left shows platelets entering the PVL channels of Anatomy A in both systolic phase and diastolic phase. Right shows a pathline of platelet from (**A**) entering the channel in systole, (**B**) reentering in diastole and recirculation for another cycle, (**C**) exiting in 3rd systole into the aorta, and finally (**D**) washing through the PVL channel in the final diastolic phase.

**Figure 4 bioengineering-10-00188-f004:**
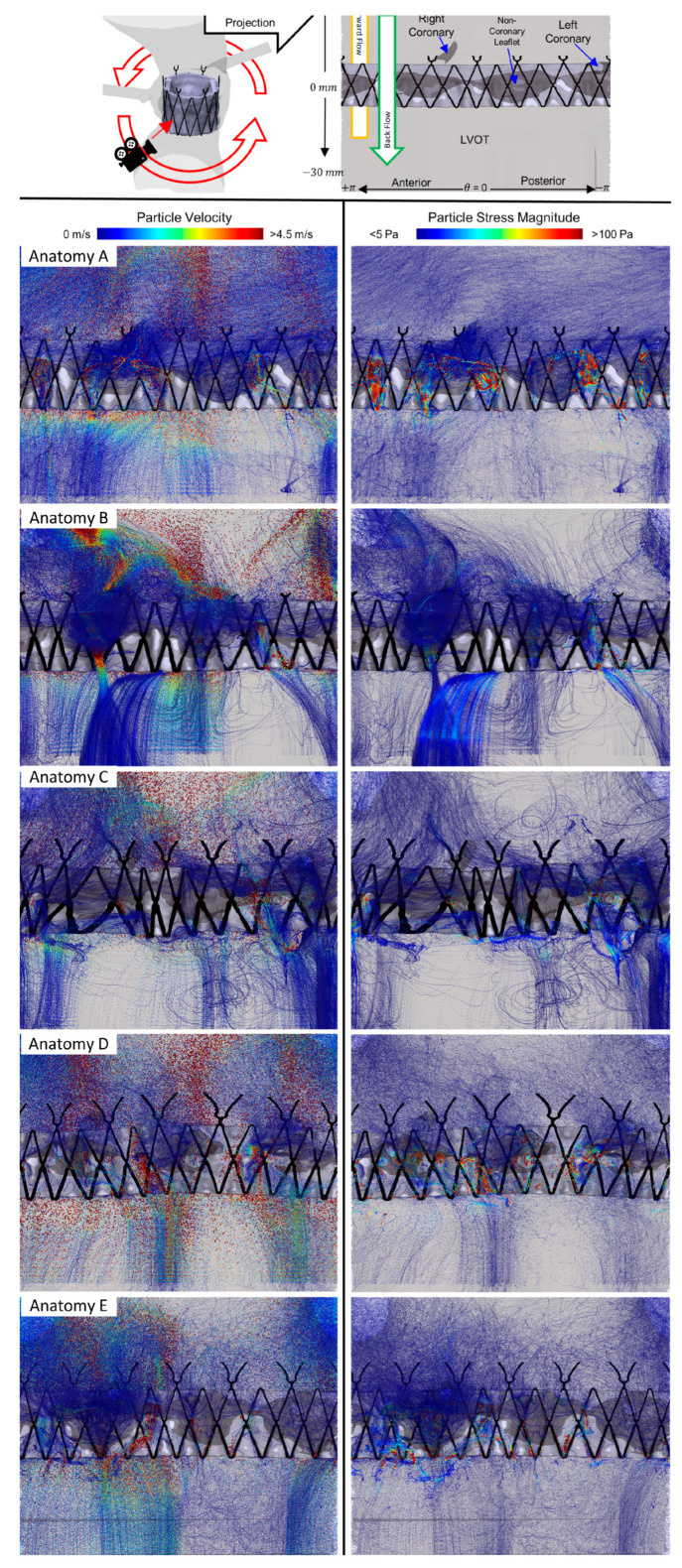
Polar projections of pathlines of the top 3000 highest SA platelets—Top—key to visualize the polar projection with red arrow demonstrating projection angle. From top to bottom are each of the patient-specific anatomies A–E. Left-column platelets are colored and sized according to the velocity magnitude, while right-column platelets are colored by stress magnitude.

**Figure 5 bioengineering-10-00188-f005:**
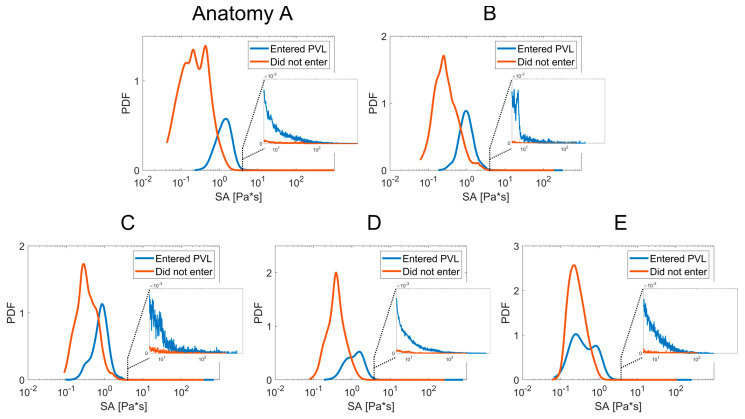
Probability density function (PDF) of the platelets stress accumulation (SA) comparison —Showing the increased stress accumulation of platelets entering the PVL channels (blue peaks shifted right). Inset graph highlights higher stress values (>5 Pa × s). Each PDF represents the patient-specific anatomies (**A**–**E**).

**Figure 6 bioengineering-10-00188-f006:**
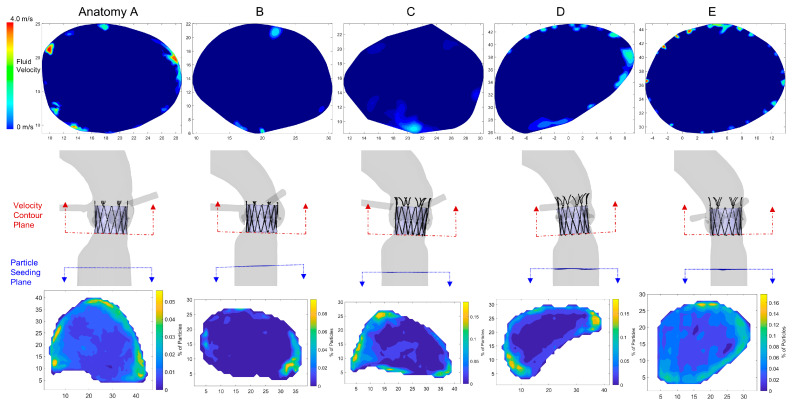
Contour plots of LVOT velocity and concentration of platelets entering the PVL domain—Top are velocity contour plots in the LVOT perpendicular to the bottom of the TAVR stent. Bottom contours represent the concentration of the starting position of the PVL platelets in the LVOT, showing concentrations towards the lumen surface. Each column represents the patient-specific anatomies (**A**–**E**).

**Figure 7 bioengineering-10-00188-f007:**
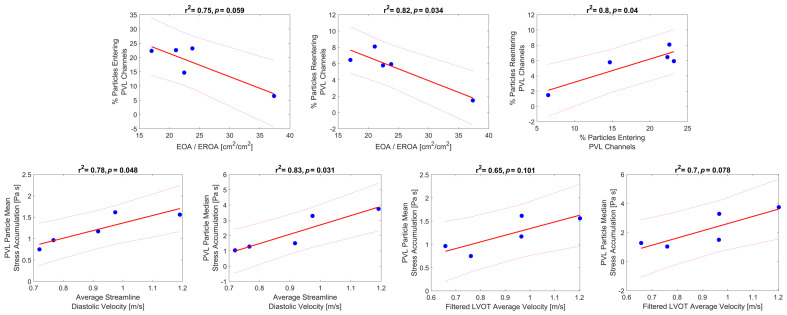
Emerging linear trends from the 5 cases—Top row—linear correlations showing the tendency of platelets to enter or reenter the PVL channels. Bottom row—linear correlations of the median and mean stress accumulation of the PVL platelets to the diastolic streamline velocity and the average velocity (>0.3 m/s) in a transverse plane of the LVOT.

**Table 1 bioengineering-10-00188-t001:** Compilation and comparison of the in vitro flow parameters to the in silico results and parameters.

	Anatomy		A	B	C	D	E	
In Vitro	CO [L/min]		4.71	4.90	4.75	5.20	4.94	*t*-Test (two-tail, equal variance)
	SV [mL/beat]		78.2	78.3	77.5	85.2	77.5	
	EOA [cm^2^]		1.39	1.08	1.03	1.17	1.48	
	Closing Flow [mL/beat]		−2.7	−3.4	−2.2	−4.0	−2.9	
	Leak Flow [mL/beat]		−7.0	−4.2	−6.7	−5.8	−3.0	
	RF [%SV]		12.4	9.7	11.5	11.5	7.6	
BCs	Avg Pressure Gradient [mmHg]	Systole	29.7	44.4	48.2	48.2	25.4	
		Diastole	−86.6	−86.8	−88.1	−88.1	-82.3	
In Silico	CO [L/min]		4.54	5.19	3.54	5.29	4.23	0.34
	SV [mL/beat]		81.2	79.4	58.2	92.2	70.0	0.61
	EOA [cm^2^]		1.50	1.05	0.80	1.17	1.22	0.58
	EROA [cm^2^]		0.067	0.028	0.034	0.069	0.058	
	Closing Flow [mL/beat]		−3.7	−1.3	−1.0	−2.0	−2.6	0.15
	Leak Flow [mL/beat]		−12.7	−3.9	−6.6	−14.6	−8.2	0.11
	RF [% SV]		20.2	6.6	13.1	18.0	15.4	0.14
	Number of Cells		3.5 M	3.5 M	2.9 M	3.4 M	3.3 M	
	Number of Platelets		2.4 M	1.9 M	1.7 M	3.3 M	2.6 M	
	% of Particle Entering PVL		14.7	6.5	23.2	22.3	22.6	
	% of Particle Re-Entering PVL		5.8	1.5	5.9	6.5	8.1	

## Data Availability

Data is contained within the article or Appendix A.

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
