# Peer review of "Mild Paravalvular Leak May Pose an Increased Thrombogenic Risk in Transcatheter Aortic Valve Replacement (TAVR) Patients-Insights from Patient Specific In Vitro and In Silico Studies"

_bioengineering, 2023, doi:10.3390/bioengineering10020188_

Round 1

Reviewer 1 Report

Authors should be congratulated for their work and their manuscript. An increased thrombogenic risk in case of PVL after TAVR is critical and diagnosis/treatment are currently debated in all trials. This article can also be used as a foundation paper for future studies for valve design. Methods are adequately described and scientific soundness is high. Conclusions are supported by results. Limitations are adequately stated.

Minor comment: please revise references. "Error! Reference source not found" is shown instead of the reference number.

Author Response

The authors would like to thank the reviewer for his appreciation of our study and its possible impact to the future of TAVR trials and devices.

Minor comment: please revise references. "Error! Reference source not found" is shown instead of the reference number.

The broken references errors were fixed in the revised submission.  This error was generated when converting to the MDPI template resulting with the link to the figure caption lost.  This was corrected.

Reviewer 2 Report

Authors of manuscript  “Mild paravalvular leak may pose an increased thrombogenic risk in transcatheter aortic valve replacement (TAVR) patients -Insights from patient specific in vitro and in silico studies” discuss an important issue regarding thrombotic risk in patients after the TAVR procedure.  In silico

and in vitro analysis of a cohort of TAVR patients classified as having mild PVL showed , that in defined CAVD anatomies platelets entering the PVL channels during systole may also re-enter the PVL channels multiple times in successive cardiac cycles. This phenomenon could significantly increase the thrombogenic risk even in mild PVL cases.

Article is well written and important from clinical point of view.

Comments:

The authors state that the thrombotic risk in patients with even a small PVL may be high, but they do not provide specific reports from the literature confirming such a risk. It would be advisable to present the percentage of thrombosis in previous studies.

 It is advisable to improve the references to the literature - Error! Reference source not found

Author Response

We would like to thank the reviewer for the kind comments and the appreciation of the significance of our study.  We believe that the phenomenon of multiple reentry into the PVL flow channels is critical for understanding how to reduce the thrombogenic risk in future devices.

Comments:

The authors state that the thrombotic risk in patients with even a small PVL may be high, but they do not provide specific reports from the literature confirming such a risk. It would be advisable to present the percentage of thrombosis in previous studies.

The authors agree with the reviewer’s comment. Unfortunately, there are limited studies relating the severity of PVL and rate of thrombosis events.  We have expanded the introduction section to include more relevant studies and rates of thrombosis in TAVR patients. Our study serves to draw the attention for tracking the presence of thrombosis of what is otherwise considered as a mild PVL and not necessarily carrying a thrombotic risk. This risk should be carefully investigated in future trials.

Edited introduction section:

“With approval of TAVR in low risk, younger, and now also bicuspid aortic valve (BAV) patients, TAVR is rapidly becoming the standard therapy to treat AS despite numerous persistent clinical complications.  TAVR is prone to various clinical complications including cardiac conduction abnormalities (CCA), poor TAVR performance due to patient-prothesis mismatch (PPM), and leakage flows between the prothesis sleeve/skirt and the lumen termed “paravalvular leak.”  These complications have been reduced in severity and prevalence with successive generations of device designs and increased experience of interventionalists, however, thrombosis and thromboembolic events remain persistent.  Major stroke rates remain between 1-5.5% [3] in newer generation devices which is reduces from the 7.8% (1 year) rates of the early PARTNER-B trial [4]. Thrombosis, which is often hypothesized to be a result of flow stagnation and unfavorable materials, leads to subclinical leaflet thickening, where the deposition of thrombosis on the aortic leaflet surface causes a malfunctioning of the prothesis.  Rates of leaflet thrombosis are greatly varied with each device trial with rates common rates between 10-15% and some studies reporting rates up to 40% of patients [5,6].  This discrepancy may be due to the lack of symptomatic or impact on patient outcomes [7] leading to reduced detection rates. With the extension of TAVR to younger patients, rates of subclinical leaflet thickening have been increasing [8-10]. Many studies [5,6] have shown rates of leaflet thrombosis are linked to unfavorable TAVR deployment parameters such as eccentric deployments [11] or reduced valve performance due to heavy patient calcification [12], as well as anatomical features such as large sinus of Valsalva leading to increased stagnation [13].

While unfavorable hemodynamics due to stagnation or material surface properties may increase the thrombogenic potential of each device, the risk of thrombosis and stroke due to PVL has not been studied rigorously and investigations into a possible link has largely been overlooked in clinical trials. PVL leak channels are complex and highly restricted flow paths due to incomplete sealing between the expanded TAVR device and underlying calcified leaflets and the aortic wall that are driven by large diastolic pressure gradients, creating high velocity jet flows from the native sinuses back into the left ventricular outflow track (LVOT). PVL is often classified by leak severity determined by clinician judgement of the jet velocity and flow, with newer generation device improvements significantly reducing severe PVL rates in clinical trials (Moderate/Severe PVL at 30 days <3.5% of patients [14], <0.8% [15]). Mild and trace PVL rates remain common with, for example, rates of no notable regurgitation at 30 days in 19.7% of patients in a recent low risk trial [14].  PVL severity is often shown to impact many post-operative outcomes and increase the patient mortality rates [16,17], which can be attributed to continued cardiac burden of often high-risk patients. An abstract by Rahgozar et al. showed no link between major stroke rates and classification of PVL [18], however larger studies have contradicted these findings.  Padang et al. [19] demonstrated that mild and tract PVL had a lower survival rate (50.9%) compared to no PVL (62.7%) at 5 years. In a recent study, Saito et al. [20] showed that mild or greater PVL had significantly lower freedom from events (70 months) compared to trace and no PVL. Additionally, PVL severity has been shown to be related to hemolysis rates [21].  

 It is advisable to improve the references to the literature - Error! Reference source not found

The broken figure references occurred when converting the manuscript to the MDPI template.  This was corrected in the revised submission.

Reviewer 3 Report

I think it's an interesting study of unique ideas.
Relationship between PVL and thrombogenic was well expressed through new models.

Author Response

We would like to thank the reviewer for taking the time to review the manuscript and the appreciation of its significance.

Reviewer 4 Report

The article presents relevant results with scientific soundness. The innovative approach using in silico models reconstructed from in vivo CT scans of patients offers a new perspective over the thrombogenic potential of paravalvular leak.

There is however, an issue with text. The following text "Error! Reference source not found.." is present several times throughout the paper. Please revise.

Author Response

The authors would like to thank the reviewer for the prompt review and appreciation of our study.  We would like to comment that the approach of µCT reconstruction of models can be expanded in the future to ex vivo samples, which further expands the possibilities for validation and verification (V&V) studies to enhance the in silico models.

There is however, an issue with text. The following text "Error! Reference source not found.." is present several times throughout the paper. Please revise.

We apologize for this error.  It was only noticed after downloading the manuscript from the submission and updating the references on our computers.  This error stemmed from the conversion to the MDPI template and loss of figure links.  It has been corrected in the new submission.